# CONFORMALIZED PHYSICS-INFORMED NEURAL NETWORKS

**Lena Podina**
Cheriton School of Computer Science
University of Waterloo
Waterloo, ON
lpodina@waterloo.ca

**Mahdi Torabi Rad**
MLBoost Inc.
Waterloo, ON
info@mlboost.tech

**Mohammad Kohandel**
Department of Applied Mathematics
University of Waterloo
Waterloo, ON
kohandel@waterloo.ca

## ABSTRACT

Physics-informed neural networks (PINNs) are an influential method of solving differential equations and estimating their parameters given data. However, since they make use of neural networks, they provide only a point estimate of differential equation parameters, as well as the solution at any given point, without any measure of uncertainty. Ensemble and Bayesian methods have been previously applied to quantify the uncertainty of PINNs, but these methods may require making strong assumptions on the data-generating process, and can be computationally expensive. Here, we introduce Conformalized PINNs (C-PINNs) that, without making any additional assumptions, utilize the framework of conformal prediction to quantify the uncertainty of PINNs by providing intervals that have finite-sample, distribution-free statistical validity.

## 1  INTRODUCTION

Physics-informed neural networks are an influential method developed by Raissi et al. (2019), which employs neural networks to solve a differential equation (forward problem) well as estimate its parameters (inverse problem). The differential equation (DE) may be ordinary (ODE) or partial (PDE). PINNs have been extended in many ways (e.g. UPINNs by Podina et al. (2023) for learning functions, XPINNs by Jagtap & Karniadakis (2021) for domain decomposition, and DeepONets by Lu et al. (2021) for operator learning), in order to solve increasingly complex problems. However, many of these methods produce only point estimates of the DE solution. This makes it difficult to apply these methods in any practical situation where the error of the solution needs to be quantified or bounded.

Uncertainty quantification methods for PINNs have been developed in order to fill this gap, such as Bayesian PINNs in Yang et al. (2021). This method involves modelling the uncertainty of model and DE parameters using Bayesian statistics. However, this approach has several significant drawbacks. An appropriate prior distribution on the neural network and DE parameters must be determined in advance, and in the case of model misspecification at this stage, the resulting uncertainties may not be calibrated, as shown by Fong & Holmes (2021). Secondly, as described in Yang et al. (2021), the noise distribution within the data needs to be assumed in advance. In Yang et al. (2021), a Gaussian noise model is assumed, but may not be an appropriate model in all cases.

Conformal prediction (CP), as introduced by Vovk et al. (2005), is a user-friendly framework for creating uncertainty intervals on predictions made by any machine learning model, including ones that are black-box or not inherently interpretable. Its distinctive feature lies in the ability to provide intervals that have finite-sample, distribution-free statistical validity. This implies that the method

offers explicit and non-asymptotic guarantees without relying on specific distributional or model-related assumptions. This makes conformal prediction a versatile tool that can be applied to any model, including neural networks. The most common flavor of CP methods is the Split Conformal method as outlined by Papadopoulos et al. (2002), which splits the otherwise training set into training and calibration sets; it then uses the former to train a point predictor and the latter to construct uncertainty intervals.

Here, we introduce Conformalized PINNs (C-PINNs) and show how they can quantify the uncertainty of: (1) the inferred DE solution (the forward problem) and (2) estimated DE parameters (the inverse problem). We also show how a procedure that utilizes the strong law of large numbers and an underlying property of Split Conformal methods can be used to *rigorously* check the *exact* coverage validity of C-PINN intervals.

## 2 METHODOLOGY AND RESULTS

For the forward and inverse problems, we consider the logistic growth Ordinary Differential Equation (ODE). This equation is widely used to model logistic growth as described by West & Newton (2019); Kohandel et al. (2006) (e.g. in a context where cells ($N$) are growing over time ($t$) but are limited by the nutritional value of the medium):

$$\frac{dN}{dt} = \beta N(1 - N); \quad N(0) = N_0 \tag{1}$$

In Section 2.1, we fix the value of $\beta$ and the initial condition $N_0$, and show the coverage validity of the solution intervals inferred by a Conformalized Physics-Informed Neural Network (C-PINN), for both noise-free and noisy datasets. In Section 2.2, we focus on the inverse problem and show a similar validity for parameter intervals. Model training and architectures are described in Appendix A.2.

For the forward problem only, we also consider the 1D Buckley-Leverett Equation Wang et al. (2021):

$$\begin{cases} u_t - f(u)_x = 0, x \in [-1, 1], t \in [0, 1]. \\ u(0, x) = \begin{cases} -3 & \text{if } x < 0 \\ 3 & \text{if } x > 0 \end{cases} \end{cases} \tag{2}$$

taking $f(u) = \dfrac{4u^2}{4u^2 + (1 - u)^2}$. Li & Mei (2021) showed that a continuous PINN was not able to successfully learn the solution to the equation, likely due to the discontinuous nature of the solution. To see if conformal prediction can still be used to construct intervals with valid coverage when PINNs do not approximate the solution well, we conformalize the forward problem for eq. 2 in Section 2.1.2.

### 2.1 CONFORMALIZING THE PINN SURROGATE SOLUTION (FORWARD PROBLEM)

In the following two subsections, we conformalize the forward problem for the logistic growth ODE (Section 2.1.1) and a PDE with a discontinuous solution, the 1D Buckley-Leverett equation (Section 2.1.2).

### 2.1.1 CONFORMALIZING THE LOGISTIC GROWTH ODE

First, a numerical solver is used to solve Equation 1 for the initial condition $N_0 = 0.1$ and $\beta = 0.05$ for time 0 to 150. A dataset of 150 tuples $\{t_i, N_i\}$ are generated in total. Gaussian noise (noise level 0.08 as per Podina et al. (2023)) is added to the solution. These are randomly split into holdout set of points (100), training points (25), and test points (25). Then, a single continuous PINN is fit to the training data as per Raissi et al. (2019), with the parameter value of $\beta$ given to the PINN in advance. After solving the forward problem, the PINN returns a surrogate solution to the differential

equation. Since the solution is modelled using a neural network, it can be evaluated at any time, not just at the training time.

We implement split conformal prediction as per Papadopoulos et al. (2002) on the solution generated by PINNs (Fig 1a). Given a prespecified miscoverage level $\alpha \in [0,1]$, this procedure creates a $(1 - \alpha)$-confidence interval for a new test point $t_{test}$ which will contain the true label $y_{test}$ with probability $P \in [1 - \alpha, 1 - \alpha + \frac{1}{(n+1)}]$ as derived in Angelopoulos & Bates (2021). This means the constructed interval has *valid coverage*. The 90% and 50% confidence intervals (with miscoverage levels $\alpha = 0.1, 0.5$ respectively) seen in the inset of Fig 1a are guaranteed to contain at least 90% and 50% of the true labels respectively.

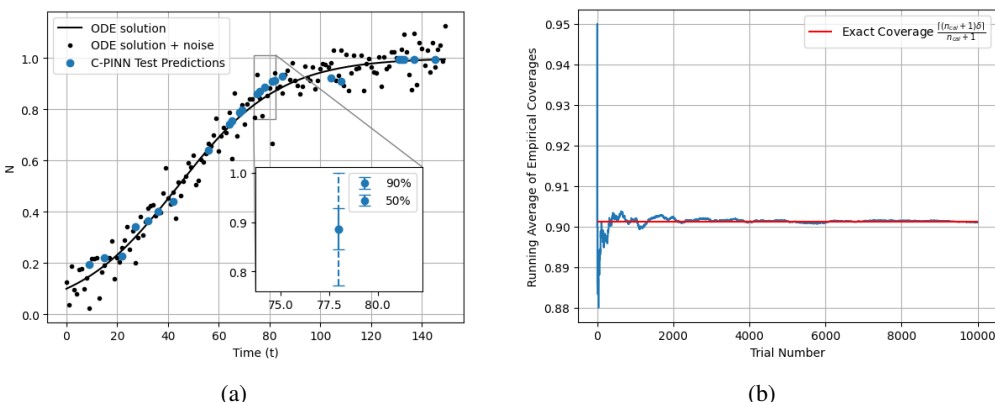

(a)                                                                 (b)

Figure 1: (a) The performance of a single PINN on noisy data (noise level 0.08). (b) Using the PINN and noisy data from Fig 1a, coverage estimated through repeated construction of an interval and validation, for different calibration and validation sets.

In order to create a prediction interval (of confidence $1 - \alpha$) for a new test point $t_{test}$, we split the holdout set of 100 tuples $\{t_i, N_i\}$ into $n_c = 80$ calibration points and $n_v = 20$ validation points. We then evaluate a non-conformity score for each calibration point. To create the scores, for each tuple $(t_i, N_i)$ of the calibration points, we evaluate the absolute error between each true $N_i$, and the corresponding predicted $\hat{N}_i$. With miscoverage value $\alpha = 0.1$ (and corresponding confidence level 90%), we calculate the $\hat{q}_\alpha = \lceil (1 - \alpha)(n_c + 1) \rceil / n_c$th quantile of the calibration non-conformity scores. For a new prediction $\hat{y}_{test}$, we can use the interval $\hat{y}_{test} \pm \hat{q}_\alpha$ as the 90% confidence interval. Any interval generated this way via split conformal prediction is proven to have valid coverage as per Angelopoulos & Bates (2021).

We then empirically show that the 90% interval has valid coverage (Fig 1b). To do this, we perform split conformal prediction over many different splits of the validation and calibration points. This procedure is outlined in Angelopoulos & Bates (2021). After constructing the interval using the calibration set, we check how many of the validation points in fact fall in the constructed interval. Angelopoulos & Bates (2021) also shows that for $n_c$ calibration points and miscoverage level $\alpha$, the coverage of the split conformal interval is exactly $\lceil (1 - \alpha)(n_c + 1) \rceil / (n_c + 1)$. Fig 1b shows that the mean coverage converges to exactly this coverage over 10000 random splits (trials), which implies that the constructed intervals are valid.

Hence, conformal prediction can be used to quantify the uncertainty and fit of a PINN solution even when there is only a single dataset available (with significant noise). Furthermore, the method yields valid coverage, which we have verified rigorously. We repeat this procedure for noiseless data, showing the fit and coverage convergence in Fig 5. The predicted 90% and 50% confidence intervals are much smaller since the PINN is better able to predict $N_{test}$ on an unseen test point $t_{test}$. However, we still have valid coverage as can be seen in Fig 5b. Hence, this method can be used to quantify the performance of the PINN on unseen test data, which is a good measure of the generalization of the solution.

### 2.1.2 CONFORMALIZING A PDE WITH A DISCONTINOUS SOLUTION (BUCKLEY-LEVERETT EQUATION)

We conformalize the forward problem as before for the Buckley-Leverett equation. A PINN is used to solve eq. 2 with 25 noiseless training points for $t = 1$. The resulting solution matches the training data, but does not adequately represent the solution near discontinuities at $x = -2.5, 3.0$. Fig 2a shows the true and predicted solution, along with the 90% and 85% confidence intervals. However, since the model only performs poorly at discontinuities, adaptive intervals would be more informative. The coverage is still as expected despite the PINN's poor performance in recovering the true solution (Fig 2b).

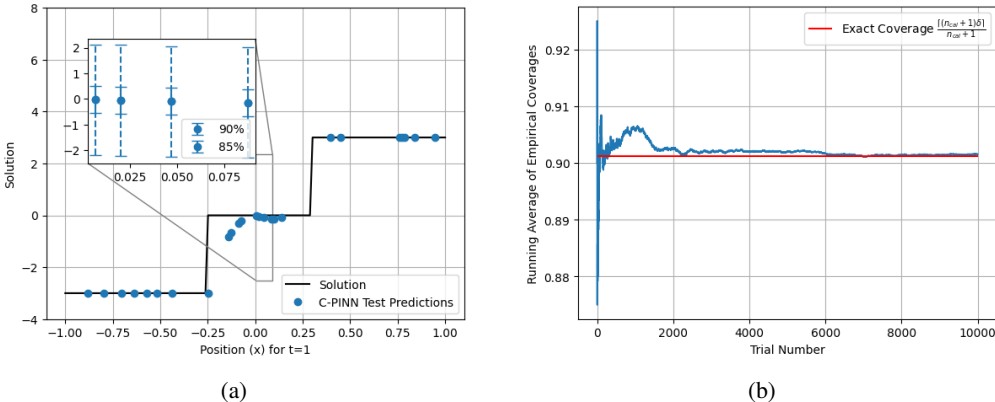

(a)                                    (b)

Figure 2: (a) The performance of a single PINN in approximating the Buckley-Leverett equation (eq. 2). (b) Using the PINN and data from Fig 2a, coverage estimated through repeated construction of an interval and validation, for different calibration and validation sets.

### 2.2 CONFORMALIZING AN ODE PARAMETER ESTIMATE (INVERSE PROBLEM)

In this section, we conformalize the inverse problem. Suppose that we are tasked with providing a valid interval over $\hat{\beta}_{test}$ for a new dataset $D_{test}$, given a distribution over possible datasets $D_i$. In the simplest form, which we explore here, the given distribution is a distribution over possible $\beta$. Then, the C-PINN can be used to construct a valid interval over the estimate $\hat{\beta}_{test}$ for $D_{test}$. We outline the procedure to do this, and show that the method has valid coverage.

To conformalize a new $\hat{\beta}_{test}$ estimate, we generate holdout pairs $\{\beta_j, D_j, \hat{\beta}_j\}$ (sampled true $\beta_j$, generated datasets $D_j$ using $\beta_j$, inferred $\hat{\beta}_j$ of $D_j$), shown in Fig 3a. We sample $\beta$ 1000 times from the given distribution, which we assume is Uniform$[0, 0.5]$. We then solve Eq 1 via an ODE solver for these values of $\beta$, generating 10 points $\{t_i, N_i\}$ per dataset. We obtain a set of 1000 tuples $\{\beta_j, D_j = \{t_i, N_i\}_j\}$. Then, for each $D_j$, we employ a standard continuous PINN to estimate the parameter $\beta_j$ in Eq 1. This yields one $\hat{\beta}$ estimate per dataset. Since each PINN can only infer $\beta$ after being trained on a specific dataset, we must use a separately initialized PINN to find $\hat{\beta}$ for each dataset. We explore alternative model options in the Conclusion section.

We perform split conformal prediction using the previously generated holdout set. For a new dataset $D_{test}$, we first use a PINN to estimate $\hat{\beta}_{test}$ for the dataset $D_{test}$. Then, we sample 800 points from the holdout set $\{\beta_j, \hat{\beta}_j\}$ to form the calibration set, and construct an 80% confidence interval as per Section 2.1. The interval given by the C-PINN will have valid coverage as long as $\beta_{test}$ in fact comes from the given distribution (Uniform$[0, 0.5]$ in this case).

The mean interval coverage converges to the expected coverage (Fig 3b) over many validation points $(\beta_{test}, D_{test})$. To compute the coverage, we repeatedly sample a new $\{\beta_{test}, D_{test}\}$ with $\beta \sim$ Uniform$[0, 0.5]$ and compute the proportion of the time that $\beta_{test}$ falls in the constructed interval. We note that for a smaller holdout set, different confidence level, and for a different distribution of $\beta$, the coverage would still be valid due to exchangeability, as discussed in Angelopoulos & Bates (2021). We repeat this experiment with noisy data (Fig 6a, 6b) and obtain similarly valid coverage,

since each dataset $D_i$ is still sampled i.i.d. Additionally, we obtain valid coverage upon repeatedly splitting the holdout set into validation and calibration points, as per Section 2.1 (Figure 7 and 8 in Appendix A.4).

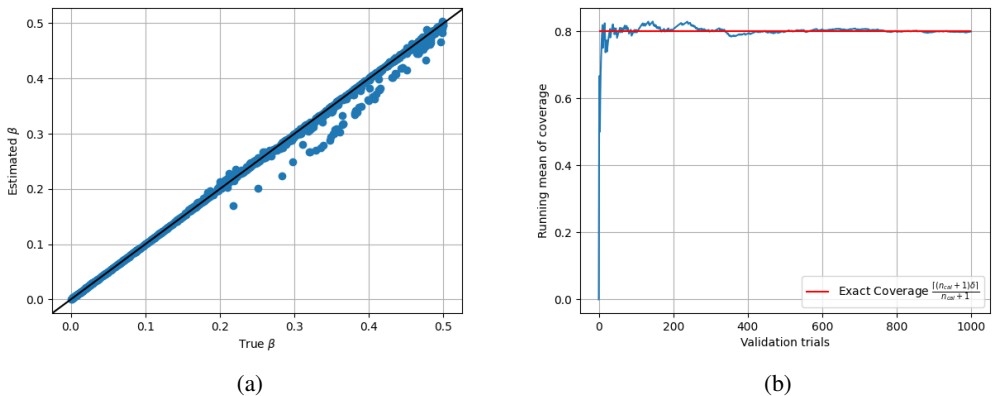

(a)                                                    (b)

Figure 3: (a) Plot of the inferred $\hat{\beta}$ vs. true $\beta$ for 1000 datasets generated from sampled values of $\beta$ (noiseless data) (b) Estimated coverage using the holdout set of $\{\beta, \hat{\beta}\}$ from Fig 3a.

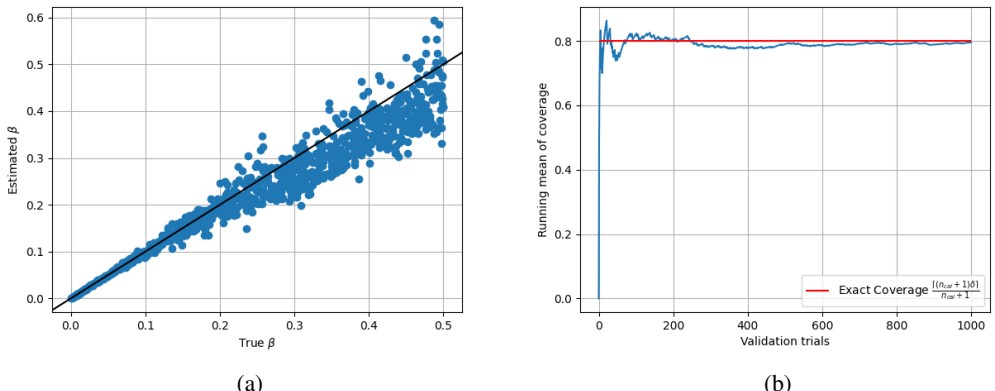

(a)                                                    (b)

Figure 4: (a) Plot of the inferred vs. true $\beta$ for 1000 datasets generated from sampled values of $\beta$ (data noise level is 0.03) (b) Estimated coverage using the holdout set of $\{\beta, \hat{\beta}\}$ from Fig 4a.

## 3   CONCLUSION AND DISCUSSION

In this work, we conformalize physics-informed neural networks (PINNs) in two ways: first, we conformalize the surrogate solution of the logistic growth ODE and the Buckley-Leverett PDE (Eqs. 1 and 2, forward problem) using a holdout calibration set and split conformal prediction as per Papadopoulos et al. (2002). In order to validate the method, we employ an averaging approach to compute the coverage of the intervals over many calibration/validation splits. Even when the data is significantly noisy, we show that the empirical coverage converges to the the theoretical coverage. Secondly, we conformalize one inferred parameter (Eq 1, inverse problem). We use the parameter estimates from many PINNs to conformalize the parameter estimate for a new dataset. The resulting interval provides valid coverage for a new, unseen value of $\beta$. In order to reduce the computational cost of this method, we note that it is possible to use UPINNs as introduced by Podina et al. (2023) to infer $\beta$ for several datasets simultaneously. Further extensions to this work include creating adaptive conformal prediction intervals for PINNs, which would tailor the interval size based on the input to the model. Another possibility is to conformalize UPINNs using split conformal prediction.

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

# A   APPENDIX

## A.1   CONFORMALIZED FORWARD PROBLEM: NOISELESS DATA

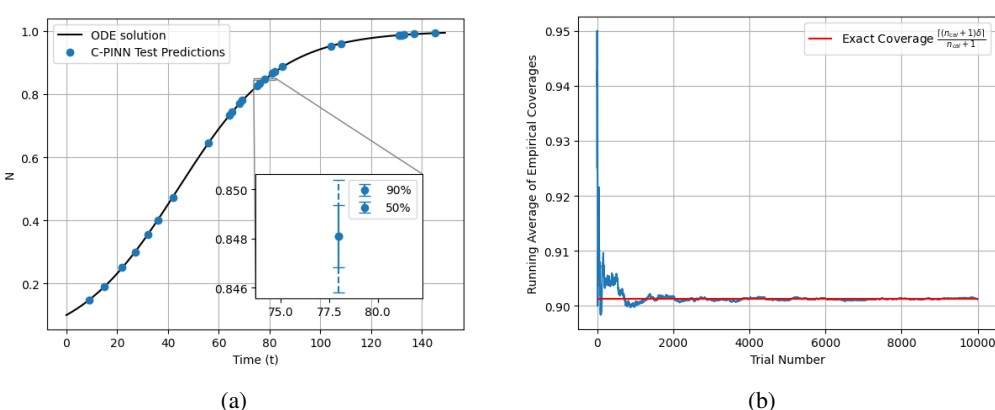

(a)                                               (b)

Figure 5: (a) The performance of a single PINN on non-noisy data. (b) Using the PINN and noiseless data from Fig 5a, coverage estimated through repeated construction of an interval, for different calibration and validation sets. The running average coverage is computed over 10000 trials.

## A.2   MODEL ARCHITECTURE AND TRAINING

The physics-informed neural network was implemented in PyTorch (Paszke et al. (2019)) similarly to https://github.com/jayroxis/PINNs. The neural network that was used to model the surrogate solution contains one input, 2 hidden layers with 10 hidden units each, and one output. The activation function used between layers is tanh. In order to train the PINN, the Adam (Kingma & Ba (2014)) optimizer (with default parameters and linear annealing learning rate scheduling) was run for 100 epochs, and then the L-BFGS optimizer was run until convergence.

## A.3   CONFORMALIZED INVERSE PROBLEM: NOISY DATA

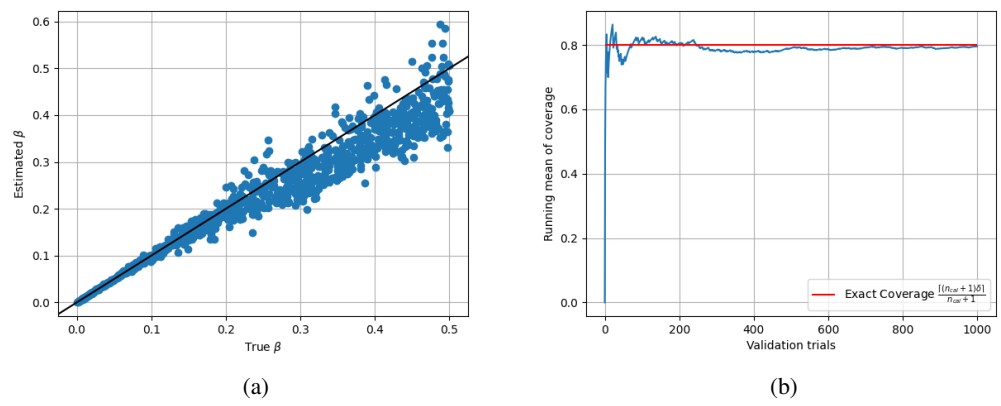

(a)                                               (b)

Figure 6: (a) Plot of the inferred vs. true $\beta$ for 1000 datasets generated from sampled values of $\beta$ (data noise level is 0.03) (b) Estimated coverage using the holdout set of $\{\beta, \hat{\beta}\}$ from Fig 6a.

## A.4 CONFORMALIZED INVERSE PROBLEM: REPEATED VALIDATION/CALIBRATION SPLITS

When conformalizing the inverse problem, in addition to testing our method on repeated i.i.d. validation samples as described in Section 2.2, we also show valid coverage using the averaging method described in Section 2.1.

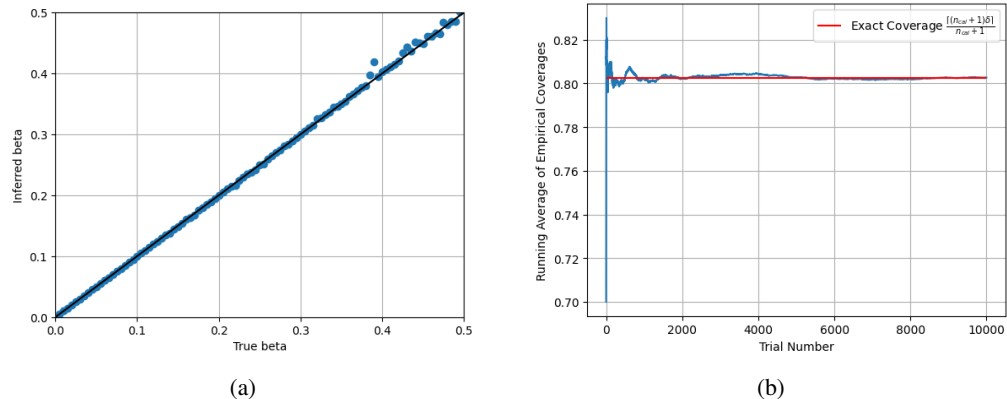

(a)                                                             (b)

Figure 7: (a) Plot of the inferred vs. true $\beta$ for 100 different values of $\beta$, equispaced from 0 to 0.5. The data is noiseless. (b) Using the $\beta$ estimates from noiseless data from Fig 7a, coverage estimated through repeated construction of an interval, for different calibration and validation sets. The running average coverage is computed over 10000 trials.

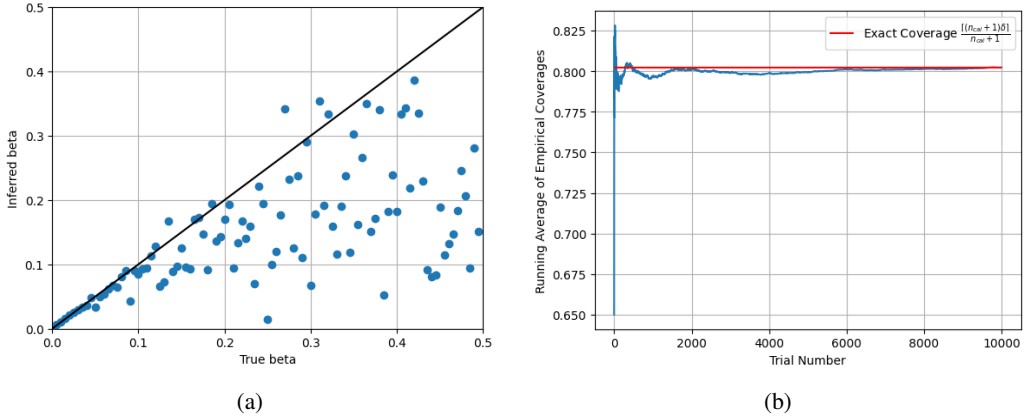

(a)                                                             (b)

Figure 8: (a) Plot of the inferred vs. true $\beta$ for 100 different values of $\beta$, equispaced from 0 to 0.5. The data is noisy with noise level 0.08. (b) Using the $\beta$ estimates and noisy data from Fig 8a, coverage estimated through repeated construction of an interval, for different calibration and validation sets. The running average coverage is computed over 10000 trials.

