# OpenReview forum: "Conformalized Physics-Informed Neural Networks"
_ICLR.cc/2024/Workshop/AI4DiffEqtnsInSci — AI4DiffEqtnsInSci @ ICLR 2024 Poster_

### Official Review · Reviewer_3JCm · 2024-02-21
**Review on Conformalized Physics-Informed Neural Neworks**

**Rating:** 7
**Confidence:** 3

**Review:**

The paper proposes using conformal prediction in the context of physics-informed neural networks (PINNs) for uncertainty quantification.

It is clearly written and well structured, and the results seem novel.

My biggest concern / question is regarding the computational cost of the method: The first experiment is made on the very simple logistic ODE, but PINNs are typically used in the context of PDEs. Is there something holding back the method from being applied to more complicated problems? This is not explicitly discussed in the paper, but it would be helpful so that readers can properly assess the utility of the proposed method. Additionally, it is good to see that the method can also be used for uncertainty quantification in parameter inference, but to the best of my understanding the proposed algorihtm does not seem very practival, as it requires many simulations, and training many different PINNs for the individual tasks; but this is acknowledged in the conclusion.

Overall, I think the paper is a good contribution to the topic of uncertainty quantification for PINNs and I therefore recommend its acceptance.

---

### Official Review · Reviewer_TMX9 · 2024-02-24
**decent work**

**Rating:** 8
**Confidence:** 2

**Review:**

This paper addresses uncertainty in PINNs through two main approaches:

1. conformalize the surrogate solution of the logistic growth ODE, ensuring that the empirical coverage aligns with theoretical expectations.
2. conformalize the inferred parameters, utilizing parameter estimates from multiple PINNs to establish valid coverage intervals for new datasets, and for unseen values of β.

Suggestions for Improvement:
1. Experimentation with more complex ODE/PDE models could enhance the method's validation.
2. The current study primarily relies on empirical evidence, lacking strong theoretical underpinnings.

---

### Meta-Review · Area_Chair_iQDe · 2024-02-25

**Recommendation:** Accept (Poster)

**Metareview:**

This paper uses conformal prediction within PINNs for uncertainty quantification, which is important in scientific applications. Both reviewers make it clear that this paper is a clear acceptance and so I also vote for acceptance.

---

### Decision · Program_Chairs · 2024-02-28

Accept (Poster)